**Data Availability Statement:** Due to confidentiality concerns, there are legal restrictions on sharing the data set for this study. The data underlying the results presented in the study are available upon

# Profiles and outcomes in patients with COVID-19 admitted to wards of a French oncohematological hospital: A clustering approach

Louise Bondeelle[1], Sylvie Chevret[2,3], Stéphane Cassonnet[3], Stéphanie Harel[4], Blandine Denis[5], Nathalie de Castro[5], Anne Bergeron 🔾[1,2]*, on behalf of The Saint Louis CORE Team Group[¶]

1 Université de Paris, Hôpital Saint-Louis, AP-HP, Service de Pneumologie, Paris, France, 2 ECSTRRA Team, Université de Paris, Inserm, Paris, France, 3 Service de Biostatistique et Information Médicale, Hôpital Saint-Louis, AP-HP, Paris, France, 4 Hôpital Saint-Louis, AP-HP, Service d'Immuno-hématologie, Paris, France, 5 Hôpital Saint-Louis, AP-HP, Service de Maladies Infectieuses, Paris, France

¶ Membership of The Saint Louis CORE Team Group is listed in the Acknowledgments.
* anne.bergeron-lafaurie@aphp.fr

## Abstract

### Objectives

Although some prognostic factors for COVID-19 were consistently identified across the studies, differences were found for other factors that could be due to the characteristics of the study populations and the variables incorporated into the statistical model. We aimed to a priori identify specific patient profiles and then assess their association with the outcomes in COVID-19 patients with respiratory symptoms admitted specifically to hospital wards.

### Methods

We conducted a retrospective single-center study from February 2020 to April 2020. A non-supervised cluster analysis was first used to detect patient profiles based on characteristics at admission of 220 consecutive patients admitted to our institution. Then, we assessed the prognostic value using Cox regression analyses to predict survival.

### Results

Three clusters were identified, with 47 patients in cluster 1, 87 in cluster 2, and 86 in cluster 3; the presentation of the patients differed among the clusters. Cluster 1 mostly included sexagenarian patients with active malignancies who were admitted early after the onset of COVID-19. Cluster 2 included the oldest patients, who were generally overweight and had hypertension and renal insufficiency, while cluster 3 included the youngest patients, who had gastrointestinal symptoms and delayed admission. Sixty-day survival rates were 74.3%, 50.6% and 96.5% in clusters 1, 2, and 3, respectively. This was confirmed by the multivariable Cox analyses that showed the prognostic value of these patterns.

request from the biostatistical department. Requests can be sent to Prof Matthieu Resche-Rigon, matthieu.resche-rigon@u-paris.fr.

**Funding:** The authors received no specific funding for this work.

**Competing interests:** The authors have declared that no competing interests exist.

## Conclusion

The cluster approach seems appropriate and pragmatic for the early identification of patient profiles that could help physicians segregate patients according to their prognosis.

## Introduction

The coronavirus disease 2019 (COVID-19) epidemic has been spreading worldwide since the beginning of 2020. Copious data concerning the clinical presentation and prognosis of the disease have been published. The results report variable mortality rates ranging from 3.2 to 28% [1] and different risk factors associated with mortality, which is predominantly secondary to respiratory failure. These differences vary depending on the geographical location of the study [2], the characteristics of the study population, including whether patients were admitted to wards and/or intensive care units (ICU), and the prognostic variables selected for inclusion in the statistical model. The difficulties regarding the prediction of the progression of COVID-19 may be attributed to the low precision of the available tools and the absence of a more global approach to prognostic prediction. At a time when countries are facing subsequent waves of the pandemic, we need complementary data and complementary statistical approaches to better classify and manage patients admitted to the hospital for COVID-19.

Many studies have focused on one predictor of mortality or ICU admission, such as lymphopenia [3], the platelet count [4] or the level of NT-proBNP [5]. Some studies have also analyzed mortality in specific populations, such as obese patients [6], diabetic patients [7], cancer patients [8] or kidney transplant recipients [9]. Although still small, the number of COVID-19 prognostic models is increasing, but their validity and widespread use remain uncertain [10, 11]. A few studies have proposed rapid scoring systems that can be used to predict mortality in critically ill patients and non-critically ill patients. These scores included clinical parameters such as heart rate, systolic blood pressure, respiratory rate, body temperature, consciousness, Glasgow coma scale (GCS) score, level of oxygen saturation, and age [12]. Nevertheless, most studies have ignored the time to death.

In contrast, various clinical presentations of COVID-19 patients have been reported; most reports have described the prevalence of each symptom separately, either from a single cohort or from meta-analyses of previous reports [13], or they have focused on specific symptoms, such as neurological symptoms (including anosmia [13] or ocular symptoms [14]).

Based on descriptive studies focused on cancer patients [8, 15, 16] and studies that matched cancer patients to a control population [17], patients with cancer who developed COVID-19 were found to be at an elevated risk of mortality/severe disease [8, 18, 19]. However, although the univariate analyses from the largest descriptive studies identified cancer as a prognostic factor, it was not confirmed by multivariate analyses [20]. One explanation could be that these studies included only a small number of cancer patients (1 to 6%) [20]. Furthermore, these studies mainly included patients with solid cancers, and few hematological malignancies [15, 21].

By contrast, less focus has been placed on patient profiles involving the entire combination of symptoms, especially in ICU, or biological measurements such as cytokine levels and their relationships with the outcome [22–24].

Herein, we aimed to a priori identify specific patient profiles of COVID-19 and their association with outcomes in a French cohort of consecutive patients with respiratory symptoms at admission to wards in our institution specializing in onco-hematology.

## Patients and methods

### Study population

We conducted a retrospective single-center study from February 2020 to April 2020. All consecutive patients who were admitted for at least 48 hours to one of the four different COVID-19 wards of Saint Louis Hospital (Paris, France), excluding those directly admitted to the ICU, were considered for inclusion in this study. Only patients with respiratory symptoms, namely, dyspnea, cough, thoracic pain and/or the need for supplemental oxygen (oxygen saturation at room air $\leq$ 94%), were selected for further analyses. The Saint Louis University Hospital is a 650-bed hospital, with 330 beds dedicated to the management of oncohematology patients.

This retrospective cohort study was approved by the Institutional Review Board of the French Learned Society for Respiratory Medicine (CEPRO 2020–029). All data were fully anonymized before we accessed them. Patients provided informed written consent to have data from their medical records used in research.

### Data collection

The dataset contained records of all patients, including their basic information (record ID, age), height, weight, body mass index, type of admission (ward or ICU), comorbidities (diabetes, high blood pressure, cardiovascular diseases, kidney failure, active malignant disease (with ongoing treatment), HIV status), medical history pertaining to COVID-19 (time of symptom onset and all clinical symptoms), laboratory measures (complete blood count, electrolytes, and inflammatory markers, such as C-reactive protein (CRP), fibrinogen, ferritin, and D-dimer), radiological findings, and use of supplemental oxygen at admission. Final survival status up to 6 months after hospital discharge was collected by phone (alive or dead), and the time to death was recorded. Patient medical information was recorded from May 3, 2020.

### Outcomes

The primary outcome was overall survival (OS), measured from the time of hospital admission until the date of last follow-up or death. At each hospital admission, the risk/benefit balance with regard to ICU transfer in case of clinical deterioration was discussed collaboratively. When the decision was not in favor of resuscitation, the patient received a "do not resuscitate" (DNR) order. For patients discharged alive, information regarding their status was obtained on September 25, 2020.

**Statistical analysis.** Summary statistics, namely the medians [interquartile ranges, IQRs], or percentages, were reported.

First, we used a principal component analysis (PCA) algorithm, which is a non-supervised statistical approach to discover inherent but hidden profiles in the patient baseline data, as measured at the time of hospital admission, with plots allowing the visualization of distance between the variables and between the patients, thus facilitating our interpretation of the data. Sixteen variables were used, namely age, body mass index, high blood pressure, malignancy, acute renal failure, chronic obstructive pulmonary disease, days elapsed since disease onset, oxygen flow at baseline, body temperature, cough, dyspnea, digestive symptoms, neurological symptoms, lymphocytes count, CRP, and platelet count. All data were scaled to unit variance. We first imputed missing values as a preliminary step before performing PCA on the complete dataset. Imputation used an iterative algorithm, consisting in (i) imputing missing values with initial values such as the mean of the variable, (ii) PCA is performed on the complete dataset, (iii) it imputes the missing values with the (regularized) fitted matrix. These three steps of estimation of the parameters via PCA and imputation of the missing values are iterated until convergence [25]. In PCA plots, similar individuals (and characteristics shared by these

individuals) were represented as points and tend to groups together, while dissimilarity, on the other hand, results in distance among the points.

Then, clustering was conducted on the five computed components of the PCA (accounting for 47.1% of the total variance) using an iterative partitioning k-means method, that aims to partition the observations into k clusters in which each observation belongs to the cluster with the nearest mean (cluster centers), minimizing within-cluster variance. Initialization used the Forgy method, that randomly chooses k observations from the dataset and uses these as the initial means. The optimal number of clusters k was estimated as the most frequently selected by 30 different indices as proposed by Charrad et al [26]. Hierarchical clustering on the results of the PCA was then conducted where starting from one cluster, the algorithm splits 3 clusters depending on the similarities measured by the distance among points.

Clinical characteristics, disease presentation, and outcome were compared across the different clusters using the chi-square test or Fisher's exact test for qualitative variables, the non-parametric Kruskal-Wallis test for continuous variables, and the log-rank test for the censored outcome. The cumulative probability of OS was plotted using the Kaplan-Meier method.

Finally, prognostic analyses for OS were conducted according to the Transparent Reporting Of a Multivariable Prediction Model for Individual Prognosis of Diagnosis (TRIPOD) reporting guidelines. A Cox proportional hazards model with a stepwise selection procedure was used to select covariates based on their statistical significance ($P < .05$) from among a list of variables with prognostic relevance according to the univariable analyses or previous findings of cluster analyses. Significant covariates were confirmed by forward selection and backward elimination techniques.

All p-values were two-sided with values $< .05$ considered statistically significant.

## Results

### Patients

A total of 330 consecutive patients hospitalized with COVID-19 in our hospital were enrolled in the study (**Fig 1**). Eighty-seven patients were directly admitted to the ICU. Among the 243 COVID-19 patients admitted to the wards, 220 (91%) patients had respiratory symptoms and were selected for further analyses. **Table 1** reports the patients' characteristics at baseline. These 220 patients were admitted to 4 wards of Saint Louis Hospital; of these, 93 (42.3%) were aged > 65 years and 75 (34.1%) had an active malignant disease.

### Clustering

Unsupervised statistical learning methods were used to discover inherent but hidden patterns in the data without any a priori hypotheses. **Fig 2** displays the data on the first axes of the PCA, exhibiting the correlation of age with CRP and oxygen flow, while old patients were likely to have no GI tract symptoms, independently of having malignancy; those patients with malignancy appeared to have more frequently high body temperature levels but were less likely to present with dyspnea.

K-means was performed on the five computed components of the PCA, summing up for 47.1% of the data variance. According to the majority rule, the best number of clusters was 3. Hierarchical clustering then segregated 47 patients in cluster 1, 87 in cluster 2, and 86 in cluster 3; the three clusters differed in terms of presentation (**Table 2**). Cluster 1 mostly included sexagenarian patients with active malignancies. Cluster 2 included the oldest patients who needed supplemental oxygen and have high C-reactive protein levels. Cluster 3 included the youngest patients with digestive symptoms. Note that inclusion of anosmia did not modify those results, with anosmia correlated with digestive disorders (**S1 Fig**).

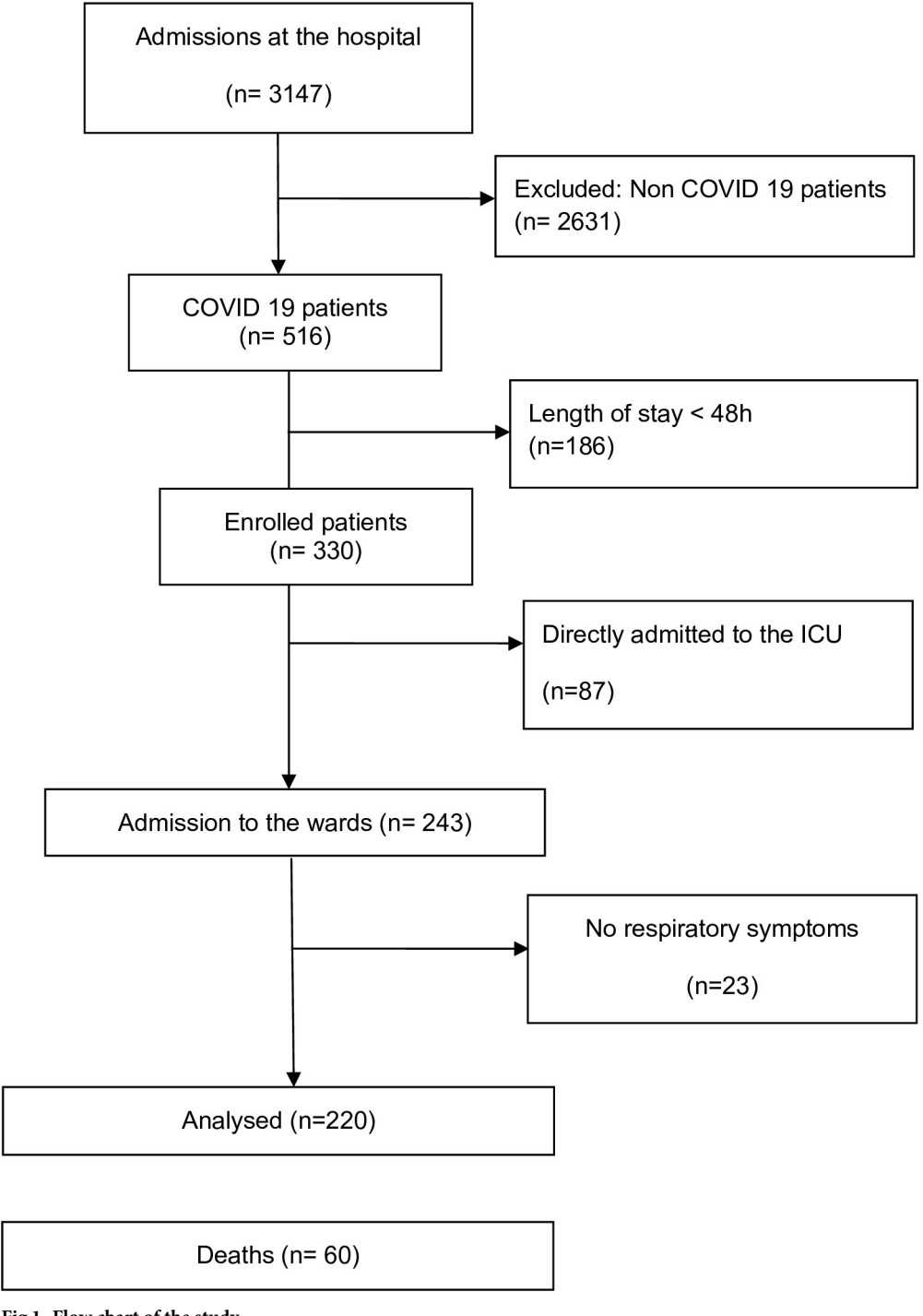

**Fig 1. Flow chart of the study.**

## Follow-up

A total of 33 patients were admitted to the ICU, 13 of whom received mechanical ventilation. Sixty patients died, 8 of whom were non-DNR patients and 50 of whom had a DNR, including 12 (25.5%) in cluster 1, 37 (42.5%) in cluster 2, and 1 (1.1%) in cluster 3. Patients with a DNR

**Table 1. Baseline characteristics of 220 patients at the time of hospital admission.**

| Characteristics | |
|---|---|
| Age, years | 62.5 [54.4–73.2] |
| Male sex | 138 (62.7%) |
| History of smoking | 59 (26.8%) |
| **Comorbidities** | |
| Body Mass Index, kg/m$^2$ | 26.8 [23.4;30.4] |
| High Blood Pressure | 105 (47.7%) |
| Cardiovascular disease* | 58 (26.4%) |
| Diabetes | 47 (21.4%) |
| Chronic respiratory disease** | 34 (15.4%) |
| Renal disease*** | 32 (14.5%) |
| Active malignant disease | 75 (34.1%) |
| Hematological disease**** | 61 (27.7%) |
| Solid tumor | 14 (6.4%) |
| **Home medication** | |
| Corticosteroids | 39 (17.7%) |
| Anticoagulant | 19 (8.6%) |
| **Disease presentation** | |
| Time to admission form first disease symptoms, days | 6 [3–9] |
| Body temperature, ˚C | 38.6 [37.8–39.1] |
| Fever | 186 (84.5%) |
| Cough | 167 (75.9%) |
| Dyspnea | 162 (73.6%) |
| Chest pain | 20 (9.3%) |
| Diarrhea | 71 (32.2%) |
| Anosmia | 21 (9.5%) |
| Neurological signs (headache, confusion) | 46 (21.1%) |
| Acute renal failure | 29 (13.2%) |
| Thromboembolic complications | 8 (3.6%) |
| Oxygen supply, L/min | 4 [2–12] |
| Lymphocytes, Giga/L | 850 [475;1220] |
| Platelets, Giga/l | 177 [105–244] |
| C Reactive Protein, mg/L | 81.5 [38.75–133.2] |
| Creatinine µmol/L | 77.5 [60;106.5] |
| DDimers (n = 64/220) | 920 [585;1702] |
| **Treatment for COVID-19** | |
| Immunosuppressive treatment***** | 42 (19.1%) |

* including 23 rythm disorders, 16 ischemic cardiopathies, 2 valvular disease, 4 others.

** asthma, chronic obstructive pneumonia disease (COPD), interstitial lung disease.

***including 12 transplanted patients, 9 dialyses (11 others).

**** including 14 non Hodgkin lymphoma, 12 acute myeloid leukemia, 7 multiple myeloma, 9 myeloid chronic leukemia, 2 acute lymphoid leukemia, 5 chronic lymphoid leukemia.

***** including 6 patients treated with Tocilizumab, 12 with dexamethasone, 9 with hydroxychloroquin, 18 with azithromycin and 17 with lopinavir/ritonavir.

order included not only cancer patients but also the oldest patients and those who had many comorbidities. Patients with a DNR order had a higher mortality rate (64.1%) than non-DNR patients (6.7%) (p-value <0.0001). The 30-day survival rate was estimated to be 75.9% (95%

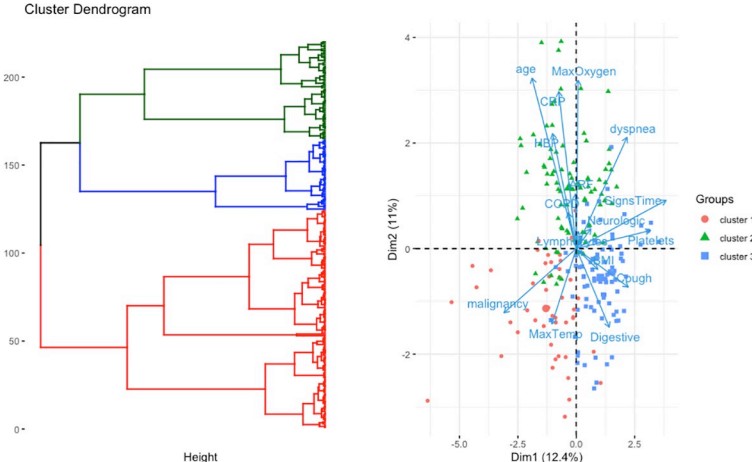

**Fig 2. Patient profiles at hospital admission.** The left panel shows the dendogram generated using the hierarchical clustering approach. The right panel shows the representation of patients and variables on the first two components of the PCA. *COPD*: *chronic obstructive pulmonary disease*, *HBP*: *high blood pressure*, *Max oxygen*: *maximum oxygen flow over 24 h of admission*, *CRP*: *C-reactive protein*, *BMI*: *body mass index; Signs Time: time since disease onset*.

CI, 70.5–81.8%) (**Fig 3**). Among the survivors, the median length of hospital stay was 5 days [IQR, 3 to 10 days], with a maximum of 56 days. We wondered whether such distinctions could have some prognostic value. Survival differed substantially across the clusters defined above, with 60-day survival rates of 74.3% (cluster 1), 50.6% (cluster 2), and 96.5% (cluster 3) (**Fig 3**).

## Prognostic analyses

Analyses of the patient outcomes were then performed in an attempt to define prognostic characteristics based on measured outcomes.

We applied a Cox proportional hazards model to predict survival among the 220 patients admitted to the wards. **Table 3** summarizes the results of the univariable and multivariable analyses. Among the 64 patients for whom the dosage was available, an elevated D-dimer level was associated with increased mortality (HR = 1.024 (IC, 1.006 to 1.042), p = 0.0084). The multivariable model identified five independent predictors of survival at the 0.001 level (**Table 3**). Interestingly, those variables were selected by non-supervised analyses and distinguished the clusters (**Table 4**).

## Discussion

In this study, we identified three clusters of patients using a non-supervised approach, that allowed data learning without any prior hypotheses. The three clusters of patients distinguished based on their initial profiles exhibited different outcomes. This was unexpected, given that such an approach only attempts to reduce the dimensionality of the dataset from information provided at baseline, thus ignoring patient outcomes. However, this finding has important implications given the emerging nature of the disease, and the need to reduce the delay in the observation of outcomes from cohort studies to understand the prognostic value of patient presentation.

Patients selected in cluster 2 had the worst survival. They were characterized by older age, more comorbidities, and a higher level of need for supplemental oxygen. Cluster 1 included a majority of patients with active malignancies and intermediate outcomes, and we identified

**Table 2. Baseline characteristics of patients at the time of hospital admission according to the patient profile defined by clustering.**

| Characteristics | Cluster 1 N = 47 | Cluster 2 N = 87 | Cluster 3 N = 86 | p-value |
|---|---|---|---|---|
| Age, | 63 [53–71] | 73 [62–82] | 55 [48–62] | <0.001 |
| years > 65 | 19 (40%) | 61 (70%) | 13 (15%) | |
| Male sex | 31 (66%) | 54 (62%) | 53 (62%) | 0.87 |
| **Comorbidities** | | | | |
| Obesity (BMI>30kg/m$^2$) | 5 (11%) | 26 (38%) | 13 (20%) | 0.002 |
| High Blood Pressure | 21 (45%) | 68 (78%) | 16 (19%) | <0.0001 |
| Cardiovascular disease | 14 (30%) | 36 (41%) | 8 (10%) | <0.0001 |
| History of smoking | 17 (36%) | 19 (23%) | 23 (28%) | 0.27 |
| Diabetes | 10 (21%) | 24 (28%) | 13 (15%) | 0.13 |
| COPD | 2 (4%) | 21 (24%) | 11 (13%) | 0.0007 |
| Chronic renal disease | 11 (23%) | 18 (21%) | 3 (3%) | <0.0001 |
| Active malignant disease | 36 (77%) | 27 (31%) | 12 (14%) | <0.0001 |
| Hematological disease | 27 (60%) | 23 (26%) | 11 (13%) | <0.0001 |
| Solid tumor | 9 (20%) | 4 (5%) | 1 (1%) | |
| **Disease presentation** | | | | |
| Time to admission from the first disease symptoms, days | 3 [1–6] | 6 [3–9] | 8 [5–13] | <0.0001 |
| Body temperature, ˚C | 38.2 [37.3;38.8] | 38 [37.45;38.7] | 38 [37.25;38.9] | 0.90 |
| Fever (Temperature > 38.5˚C) | 43 (91%) | 67 (77%) | 76 (98%) | 0.055 |
| Cough | 32 (68%) | 61 (71%) | 74 (86%) | 0.019 |
| Dyspnea | 11 (24%) | 76 (87%) | 75 (88%) | <0.0001 |
| Thoracic pain | 2 (4.3%) | 7 (8.2%) | 11 (13.1%) | 0.24 |
| Diarrhea or nausea | 12 (25%) | 13 (15%) | 46 (53%) | <0.0001 |
| Anosmia | 3 (6.4%) | 5 (5.8%) | 13 (15.1%) | 0.10 |
| Headache or confusion | 1 (2%) | 20 (23%) | 25 (29%) | 0.0002 |
| Acute renal failure | 2 (4%) | 24 (28%) | 3 (4%) | <0.0001 |
| Need for Oxygen supply | 32 (14%) | 30 (14%) | 42 (19%) | 0.0009 |
| Oxygen supply in applicants, L/min | 2 [2–3] | 3 [2–8] | 2 [2–3] | <0.0001 |
| Lymphocytes, Giga/L | 560 [275–925] | 840 [560–1205] | 1030 [647–1445] | 0.002 |
| Platelets, Giga/L | 130 [44–190] | 154 [100–224] | 211 [155–270] | <0.0001 |
| C reactive protein, mg/L | 39.5 [9.7–73.2] | 99.5 [53.0–198.5] | 65.0 [31.5–121.0] | <0.0001 |
| DDimers, µg/L | 795 [520–965]; n = 6 | 910 [620–2205]; n = 27 | 970 [560–1580]; n = 31 | 0.60 |

COPD: Chronic obstructive pulmonary disease.

specific patients managed at our institution. The characteristics of these patients may reflect the management and close follow-up provided because of the underlying active malignancy: early admission after the onset of COVID-19 symptoms and lower lymphocyte and platelet cell counts. DNR orders, which have been previously found to be a predictor of mortality [27], were less common in cluster 1 than in cluster 2. This is probably because of the selection of patients with active malignancies but few comorbidities and younger age in cluster 1. Finally, cluster 3 had the best outcome and included the youngest patients, who were characterized by relatively few comorbidities and COVID-related gastrointestinal (GI) symptoms.

The prognostic value of the clusters was confirmed by multivariable prognostic analyses, which selected five key independent clinical variables associated with mortality, namely, age, active malignancy, dyspnea, supplemental oxygen (>5 L/min), and acute renal failure, which were also differentially distributed across the clusters. These results, obtained from an

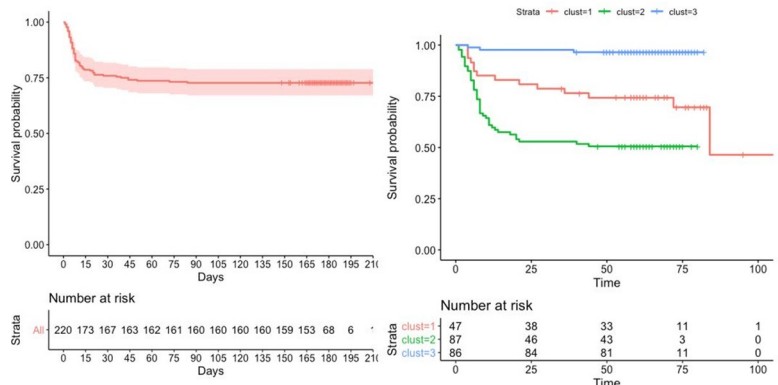

**Fig 3. Survival since hospital admission, overall (left plot) and according to the clusters (right plot).**

**Table 3. Univariable and multivariable prognostic analyses of survival (at admission in patients not directly admitted to ICU).**

| Predictor | Univariable models | | Multivariable models | |
|---|---|---|---|---|
| Predictor | HR (95%/CI) | p-value | HR (95%/CI) | p-value |
| Age, /10 years | 1.62 (1.36–1.93) | < .0001 | 1.74 (1.43–2.10) | < .0001 |
| Female sex | 0.64 (0.36–1.12) | 0.12 | | |
| Time since disease onset, weeks | 0.88 (0.78–0.98) | 0.02 | | |
| **Comorbidities** | | | | |
| Obesity, Body Mass Index >30kg/m$^2$ | 1.39 (0.77–2.49) | 0.27 | | |
| Active solid or hematological malignancy | 3.89 (2.30–6.59) | < .0001 | 7.72 (4.28–13.9) | < .0001 |
| Preexisting chronic renal failure | 2.75 (1.55–4.87) | .0006 | | |
| COPD | 1.82 (1.00–3.32) | 0.049 | | |
| Cardiovascular disease | 1.81 (1.07–3.06) | 0.027 | | |
| High Blood Pressure | 1.13 (0.68–1.87) | 0.64 | | |
| Diabetes mellitus | 1.13 (0.62–2.07) | 0.68 | | |
| **Disease presentation** | | | | |
| Fever | 0.65 (0.34–1.22) | 0.18 | | |
| Oxygen supply $\geq$ 5L/min | 2.65 (1.52–4.61) | .0005 | 2.67 (1.50–4.75) | < .0001 |
| dyspnea | 1.40 (0.76–2.59) | 0.28 | 3.75 (1.85–7 .55) | 0.0002 |
| Cough | 0.54 (0.32–0.982 | 0.02 | | |
| Diarrhea or nausea | 0.32 (0.18–0.70) | 0.0029 | | |
| Anosmia | 0.57 (0.15–1.49) | 0.20 | | |
| Headache or confusion | 1.15 (0.63–2.09) | 0.64 | | |
| Acute renal failure* | 2.31 (1.27–4.321 | 0.006 | 4.33 (2.21–8.48) | < .0001 |
| **Biology** | | | | |
| C reactive protein (mg/L) | 1.75 (1.36–2.25) | < .0001 | | |
| Platelets (G/L) /10Giga | 0.05 (0.93–0.98) | 0.0010 | | |
| Lymphocytes counts (G/L) | 0.99 (0.99–1.05) | 0.74 | | |
| DDimers (/100UI) | 1.02 (1.01–1.04) | 0.0084 | | |
| **Clusters, profiles** | | | | |
| Cluster 1 (reference) | 1.00 | | | |
| Cluster 2 | 1.03 (1.11–3.71) | 0.02 | | |
| Cluster 3 | 0.10 (0.03–0.36) | 0.0004 | | |

*defined as creatinine rate above the normal value.

Table 4. Outcomes according to patient profiles defined by clustering.

| Follow-up and Outcomes | Cluster 1 N = 47 | Cluster 2 N = 87 | Cluster 3 N = 86 | p-value |
|---|---|---|---|---|
| Maximal body temperature over the hospital stay, ˚C | 38.4 [38.0–39.3] | 38.6 [37.7–39.3] | 38.6 [38.0–39.0] | 0.86 |
| Maximal oxygen supply over the hospital stay, L/min | 2 [0–4] | 6 [3–14] | 2 [1–4] | <0.0001 |
| Do not resuscitate | 16 (35%) | 56 (74%) | 6 (8%) | <0.0001 |
| Secondary ICU admission | 1 (2%) | 16 (18%) | 16 (19%) | <0.001 |
| Intubation with mechanical ventilation | 0 | 8 (9%) | 5 (6%) | 1.00 |
| Death | 14 (30%) | 43 (49%) | 3 (3%) | <0.0001 |

unsupervised model, confirm and reinforce the results obtained with supervised analytical models [6, 12, 19, 28].

Older age is certainly the strongest risk factor for a poor outcome. It has been identified as such in most of the studies, regardless of the population studied [12]. A recent meta-analysis further identified more than 30 independent clinical or biological risk factors for severe COVID-19, most of which were in agreement with the results of previous meta-analyses [29]. Although our findings may appear somewhat expected, with young and old patients differing in terms of both presentation and outcomes, the poor outcome of patients with malignancies is worthy of attention. Indeed, surprisingly, malignancy was not associated with a poor outcome in multivariate analyses in previous studies in the general patient population. This may reflect the small proportion of patients with cancer in most studies and the lack of distinction between patients with active cancer or a past history of cancer [20, 30]. Laboratory findings have also been associated with the prognosis of COVID-19, including blood cell counts, markers of inflammation, and coagulation factors. As practices have evolved over time, we were only able to include biological parameters that have been routinely used in the cluster model, i.e., blood cell counts and the levels of CRP and creatinine. Unfortunately, while the D-dimer level was associated with prognosis in many studies and its measurement has become routine, missing data in our cohort prevented us from including this parameter in the multivariable analysis. We were also not able to study the levels of IL-6 and troponin. However, we used simple clinical and biological parameters that are accessible in all centers, making our findings pragmatic.

We found that patients with GI tract symptoms had better survival than the others, although most of them had dyspnea, which is known to be a poor prognostic factor. This finding is in accordance with recent data that further showed that patients with GI symptoms have reduced levels of circulating cytokines associated with inflammation and tissue damage [31]. This could be explained by the various settings of those studies, which involved the overall population (mostly in China) or hospitalized patients (Europe and the US) and sometimes focused on specific subpopulations, such as patients with hypertension [32].

Our study has limitations. As this was a retrospective study, there is always potential for biases. It should be noted that at the time of the study, the treatment for COVID-19 was not standardized. Nonetheless, we made systematic efforts to obtain a thorough and detailed history from each patient included in the study, in part by performing a chart review, and we performed prolonged follow-up after patient discharge. The study was monocentric, given that it was scheduled in the emergency context of the pandemic; however, patients were prospectively enrolled in 4 different wards in the hospital, involving different specialists (from pulmonary to infectious diseases, post-emergency care or internal medicine) and therefore cared for by different teams, which somewhat increases its external validity. Furthermore, the outcome results were extracted from the very early cases considered in the "first wave" of COVID-19; it would

be interesting to investigate whether it has changed since then in the following waves. Last, we used PCA as the method of data reduction, while new methods of dimensionality reduction such as autoencoders based on neural networks may have been used, that have the potential of handling non-linearity, allowing the model to learn more powerful generalizations compared to PCA, and to reconstruct the input with significantly lower information loss [33]. Other data mining techniques that combined non-supervised and supervised information, such as sub-group discovery which extracts interesting rules with respect to a target variable, or semi-supervised learning methods, could also be of interest. However, we placed ourselves in the setting of define clusters of patients from baseline information, that is, of evaluating patient profile when no target outcome could be available, even if its relationships with the outcome could be of interest.

## Conclusion

This study in a large cohort of COVID-19 patients admitted to wards with respiratory symptoms identified different patient profiles based on their history and presentation at the time of hospital admission; these profiles correlated with patient outcomes. This study emphasized the heterogeneity among the profiles and outcomes of COVID-19 patients in hospitalized wards, as well as the similarities of profiles compared to a recent Spanish cohort. The cluster approach seems appropriate and pragmatic to help physicians segregate patients according to their predicted outcomes.

## Supporting information

**S1 Fig. Patient profiles at hospital admission.** Representation of patients and variables (including anosmia) on the first two components of the PCA. *COPD*: *chronic obstructive pulmonary disease*, *HBP*: *high blood pressure*, *Max oxygen*: *maximum oxygen flow over 24 h of admission*, *CRP*: *C-reactive protein*, *BMI*: *body mass index; Signs Time*: *time since disease onset*.
(TIF)

## Acknowledgments

The Saint Louis CORE group is a collaborating group of clinicians, radiologists, biologists, pharmacists and clinical research assistants of Saint Louis Hospital. They all have participated to the care of patients with COVID-19 and/or to research into COVID-19 in Saint Louis Hospital, Paris, during the SARS-COV-2 epidemic. They decided to share their data to ease local research into COVID-19. All the manuscript written on behalf of the Saint Louis CORE group has been, preliminary to submission, sent to all members for critical rereading and consent for publication.

Members of the Saint Louis CORE group: Achili Y, Ades L, Aguinaga L, Archer G, Benattia A, Bercot B, Bertinchamp R, Bouaziz JD, Bouda D, Boutboul D, Brindel Berthon I, Bugnet E, Caillat Zucman S, Celli Lebras K, Chabert J, Chaix ML, Clément M, Davoine C, De Kerviler E, De Margerie-Mellon C, Delaugerre C, Depret F, Djaghout L Dupin C, Farge-Bancel D, Fauvaux C, Feghoul L, Feredj E, Feyeux D, Fontaine JP, Fremeaux-Bacchi V, Galicier L, Garestier J, Jegu AL Kozakiewicz E Lebel M Baye A, Le Goff J, Le Guen P, Lengline E, Liegeon G, Lorillon G, Madelaine Chambrin I, Mahjoub N, Martin de Frémont G, Maylin S, Meunier M, Molina JM, Morin F, Oksenhendler E, Peffault de la Tour R, Peyrony O, Plaud B, Rouveau M, Salmona M, Saussereau J, Schnepf N, Soret J, Tazi A, Thegat M, Tremorin MT.

## Author Contributions

**Conceptualization:** Louise Bondeelle, Sylvie Chevret, Anne Bergeron.

**Data curation:** Louise Bondeelle, Stéphane Cassonnet, Stéphanie Harel, Blandine Denis, Nathalie de Castro.

**Formal analysis:** Louise Bondeelle, Sylvie Chevret.

**Investigation:** Nathalie de Castro, Anne Bergeron.

**Methodology:** Sylvie Chevret, Anne Bergeron.

**Project administration:** Stéphane Cassonnet.

**Resources:** Stéphanie Harel, Blandine Denis, Nathalie de Castro.

**Software:** Sylvie Chevret, Stéphane Cassonnet.

**Supervision:** Sylvie Chevret, Anne Bergeron.

**Validation:** Stéphanie Harel, Blandine Denis, Nathalie de Castro, Anne Bergeron.

**Visualization:** Anne Bergeron.

**Writing – original draft:** Louise Bondeelle, Stéphanie Harel.

**Writing – review & editing:** Louise Bondeelle, Sylvie Chevret, Stéphane Cassonnet, Blandine Denis, Nathalie de Castro, Anne Bergeron.

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
