## [Decision Letter · Decision Letter 0]

2 Mar 2021

PONE-D-21-02959

Profiles and outcomes in patients with COVID-19 admitted to wards of a French oncohematological hospital: a clustering approach

PLOS ONE

Dear Dr. Anne Bergeron,

Thank you for submitting your manuscript to PLOS ONE. After careful consideration, we feel that it has merit but does not fully meet PLOS ONE’s publication criteria as it currently stands. Therefore, we invite you to submit a revised version of the manuscript that addresses the points raised during the review process.

ACADEMIC EDITOR: The reviewers have raised a number of points which we believe major modifications are necessary to improve the manuscript, taking into account the reviewers' remarks. Please consider and address each of the comments raised by the reviewers before resubmitting the manuscript. This letter should not be construed as implying acceptance, as a revised version will be subject to re-review.

We look forward to receiving your revised manuscript.

Kind regards,

Wisit Cheungpasitporn, MD

Academic Editor

PLOS ONE

Journal Requirements:

2)  In your ethics statement in the Methods section and in the online submission form, please provide additional information about the data used in your retrospective study. Specifically, please ensure that you have discussed whether all data were fully anonymized before you accessed them and/or whether the IRB or ethics committee waived the requirement for informed consent. If patients provided informed written consent to have data from their medical records used in research, please include this information.

3)  Thank you for providing the date(s) when patient medical information was initially recorded. Please also include the date(s) on which your research team accessed the databases/records to obtain the retrospective data used in your study.

4)  We note that you have indicated that data from this study are available upon request. PLOS only allows data to be available upon request if there are legal or ethical restrictions on sharing data publicly. For information on unacceptable data access restrictions, please see http://journals.plos.org/plosone/s/data-availability#loc-unacceptable-data-access-restrictions.

5) PLOS requires an ORCID iD for the corresponding author in Editorial Manager on papers submitted after December 6th, 2016. Please ensure that you have an ORCID iD and that it is validated in Editorial Manager. To do this, go to ‘Update my Information’ (in the upper left-hand corner of the main menu), and click on the Fetch/Validate link next to the ORCID field. This will take you to the ORCID site and allow you to create a new iD or authenticate a pre-existing iD in Editorial Manager. Please see the following video for instructions on linking an ORCID iD to your Editorial Manager account: https://www.youtube.com/watch?v=_xcclfuvtxQ

Reviewers' comments:

Reviewer's Responses to Questions

**Comments to the Author**

1. Is the manuscript technically sound, and do the data support the conclusions?

Reviewer #1: Partly

Reviewer #2: Partly

Reviewer #3: Partly

Reviewer #4: Partly

2. Has the statistical analysis been performed appropriately and rigorously? 

Reviewer #1: No

Reviewer #2: Yes

Reviewer #3: No

Reviewer #4: Yes

3. Have the authors made all data underlying the findings in their manuscript fully available?

Reviewer #1: No

Reviewer #2: Yes

Reviewer #3: No

Reviewer #4: Yes

4. Is the manuscript presented in an intelligible fashion and written in standard English?

Reviewer #1: Yes

Reviewer #2: Yes

Reviewer #3: Yes

Reviewer #4: Yes

5. Review Comments to the Author

Reviewer #1: In this case, I recognise the potential interest of your findings for the COVID specialists perhaps in the future. However, I regret that I cannot conclude that the paper offers the sort of advance in fundamental scientific understanding or technological capability that would be likely to excite the immediate interest of the broader readership. It does not offer any scientific advancement in a reviewer's mind on how this work is going to solve anything in such a global crisis.

As a manuscript it has very poor quality representation with only three/four figures which does not add much to the story. It's hard to accept that simple clustering is going to solve the problem of overcrowded hospital wards around the world with so many dying covid patients!

Reviewer #2: 1. Mathematical discussion on each algorithm is essential, it will make the article easy for the

reader

Reviewer #3: The paper presents a statistical analysis of 220 patients admitted to wards in a French hospital from Feb to Apr 2020. The study was carried out in a single location and not validated somewhere else.

Data used in the study are not publicly available during the time of this review. The instructions are to email somebody to request the data.

For the analysis, the authors collect a profile for every patient accepted to a ward with respiratory symptoms. A principal components analysis is applied over the profiles to identify patterns in the data. In the analysis, the authors identified 3 clusters of patients, which are then characterised in this paper.

**Main comments**

PCA and clustering:

- The main objective of this work is to classify patients according to their probability to survive COVID-19 or not. The probabilities come from an unsupervised clustering. The clustering done is very shallow and based on a "blackbox" tool, which is not described properly. It's not clear why 3 is the best number of clusters or what parameters where used during the hyper-parameter search.

- It is not clear what features where used or if any transformation was used over the raw profiles. Labels or outcomes are also not clearly explained. Did you collect this information for all patients? Where cases where no answer could be obtained about the patient?

- How do you deal with missing values in the data? For example, not all patients have an active malignant disease. How does your clustering approach deals with that?

- Data normalisation is a must when applying PCA. What normalisation was applied to the data?

- While describing the clustering used, i.e., K-means, it is not clear how many components were computed and how many were used.

- In the description of the results, it's not clear what causes of mortality are associated with each of the clusters. This brings me to think that K-means may not be the best approach to inform practitioners. Could be more useful in this case to consider a hierarchical clustering where you can identify subgroups similarity?

Impact and usability of the results:

- I agree with the authors that the results presented here are somehow expected considering the selection criteria of the cohort

- My expectation of a paper like this is to present materials and methods along with the reproducible results that can be adopted in another institution. However, to that end, there are missing details to help others to use your results.

- How can we test the reproducibility of your results? What should another hospital do to adopt your approach? What considerations should be taken? (e.g., this only works for patients above 48 years old)

Questions that require an answer:

- How does the admission criteria to wards look like? Is there any standard followed?

- Are patients with lost of the sense of smell considered in your analysis? This is based on various cases of asymptomatic patients that may not be considered in your study

- The outcome results are extracted from the very early cases considered in the "first wave". I think it would be interesting to know if this has changed since then in the following waves (second, third, etc.)

- Why the results were not compared against supervised models? I ask this because it should be very simple to perform this comparison.

- [Line 270] "the poor outcome of patients with malignancies is worthy of attention" To me this is what deserves more attention and what could bring new insights into this study.

**Minor comments**

- [Line 84] First mention of SARS-CoV-2, previously you refer only to COVID-19

- [Line 125] Typo, "Firs" → "First"

- [Line 125] "nonsupervised" → "non-supervised" Use hyphen

- [Line 133] Missing reference to the “NbClust” package

- [Line 134] Which are the best hyper-parameters found by the package?

- [Line 138] "nonparametric" → "non-parametric"

- [Line 144] p-values are denoted in different ways across the text, e.g., uppercase, lowercase, italic. Please chose one and stick to it.

- [Line 202] "Base line" → "Baseline"

- [Line 208] "mechanically" → "mechanical"

- [Line 215] "iQR" → "IQR" all in uppercase

- [Figure 1] It was not mentioned earlier in the text that patients whose stay was less than 2 days were left out of the analysis

- [Figure 3] What's the unit of time here?

Reviewer #4: In this paper, the authors analyze influence factors regarding the prognosis of a COVID-19 infection.

In general, the paper is well written and highly relevant to the current situation. However, a few points should be clarified.

- For me it is unclear how exactly the clustering is performed. In lines 131-132 you write that an iterative partitioning k-means method is used. This contradicts

lines 133-135 where you say that the best clustering result is used (obtained by trying all combinations of the number of clusters,

distance measures and clustering methods).

- In the PCA that your are doing, I think that you reduce the data-dimensions to two, this is not exactly described. What percentage of data is described by these two

components?

- In line 156: Why exactly is 65 used as the age limit here? Can you present a histogram of the age distribution here?

- The discussion of the results is rather brief and much is left to the reader to infer. Please expand the discussion of the results further.

More Points

-------------

- Line 125: Firs_t_, we use ....

Recommendations

--------------

- I would like to suggest that autoencoder is used for dimensionality reduction, because it also finds nonlinear relationships in the data.

- If you are looking for clusters that have high mortality, I would recommend the subgroup discovery method.

6. PLOS authors have the option to publish the peer review history of their article (what does this mean?). If published, this will include your full peer review and any attached files.

Reviewer #1: No

Reviewer #2: No

Reviewer #3: **Yes: **Emir Munoz

Reviewer #4: No

---

## [Author Response · Author response to Decision Letter 0]

16 Mar 2021

Review Comments to the Author

Reviewer #1: In this case, I recognise the potential interest of your findings for the COVID specialists perhaps in the future. However, I regret that I cannot conclude that the paper offers the sort of advance in fundamental scientific understanding or technological capability that would be likely to excite the immediate interest of the broader readership. It does not offer any scientific advancement in a reviewer's mind on how this work is going to solve anything in such a global crisis.

As a manuscript it has very poor quality representation with only three/four figures which does not add much to the story. It's hard to accept that simple clustering is going to solve the problem of overcrowded hospital wards around the world with so many dying covid patients!

Answer: We thank the reviewer for recognizing a potential interest in our results. Our study provides a methodological approach that is different and complementary to other available studies and as such, it seems to be relevant to other reviewers. Notably, by defining patient profiles combining clinical and biological patient characteristics measured routinely at the time of hospital admission, it exhibits their correlations and differences in disease presentation, rather than by analyzing those variables, separately; the fact that those clusters were associated with the outcome was another point of interest. We agree that our paper alone will not solve the problem of hospital overcrowding and significant mortality associated with COVID 19 but we believe that specifying patient profiles associated with different prognoses can be very useful for further management of these patients. We hope that the clarifications made in response to the other reviewers’comments will make our manuscript clearer.

Reviewer #2: 1. Mathematical discussion on each algorithm is essential, it will make the article easy for the reader

Answer: We have detailed the clustering algorithm, based on K-means clustering that aims to partition the observations into k clusters in which each observation belongs to the cluster with the nearest mean (cluster centers or cluster centroid), minimizing within-cluster variance. It is also referred to as Lloyd's algorithm, particularly in the computer science community. Initialization used the Forgy method, that randomly chooses k observations from the dataset and uses these as the initial means. Several diagnostic checks were used for determining the number of clusters in the data set, and that which is selected more frequently by these indices is retained as the optimal number of clusters as proposed by Charrad et al (2014). Thus, based on those indices, 3 was the most frequently selected number of clusters. Note that 

All these points have been more clearly stated in the revised manuscript.

Reviewer #3: The paper presents a statistical analysis of 220 patients admitted to wards in a French hospital from Feb to Apr 2020. The study was carried out in a single location and not validated somewhere else.

Data used in the study are not publicly available during the time of this review. The instructions are to email somebody to request the data.

For the analysis, the authors collect a profile for every patient accepted to a ward with respiratory symptoms. A principal components analysis is applied over the profiles to identify patterns in the data. In the analysis, the authors identified 3 clusters of patients, which are then characterised in this paper.

**Main comments**

PCA and clustering:

- The main objective of this work is to classify patients according to their probability to survive COVID-19 or not. The probabilities come from an unsupervised clustering. The clustering done is very shallow and based on a "blackbox" tool, which is not described properly. It's not clear why 3 is the best number of clusters or what parameters where used during the hyper-parameter search.

Answer: Actually, the main objectives of the paper were not to classify patients according to their survival status, but – as reported in the Abstract and the manuscript- to “a priori identify specific patient profiles” while studying their association with the outcomes was only a secondary objective.

We do not think the clustering is a black box, given it was based on K-means clustering that aims to partition the observations into k clusters in which each observation belongs to the cluster with the nearest mean (cluster centers or cluster centroid), minimizing within-cluster variance. It is also referred to as Lloyd's algorithm, particularly in the computer science community. Initialization used the Forgy method, that randomly chooses k observations from the dataset and uses these as the initial means. Several diagnostic checks were used for determining the number of clusters in the data set, and that which is selected more frequently by these indices is retained as the optimal number of clusters as proposed by Charrad et al (2014). Thus, based on those indices, 3 was the most frequently selected number of clusters, as detailed in the figure below. This has been clarified in the revised manuscript.

Reference

Charrad, Malika, Nadia Ghazzali, Véronique Boiteau, and Azam Niknafs. 2014. “NbClust: An R Package for Determining the Relevant Number of Clusters in a Data Set.” Journal of Statistical Software 61: 1–36

- It is not clear what features where used or if any transformation was used over the raw profiles. Labels or outcomes are also not clearly explained. Did you collect this information for all patients? Where cases where no answer could be obtained about the patient?

Answer: We considered 16 base line variables, namely age, body mass index, high blood pressure, malignancy, acute renal failure, chronic obstructive pulmonary disease, days elapsed since disease onset, oxygen flow at base line, body temperature, cough, dyspnea, digestive symptoms, neurological symptoms, lymphocytes count, C Reactive Protein, and platelet count. All these data were extracted from all the patient medical files, though there were some missing values (see answer to next point). 

This has been more clearly reported in the revised manuscript.

- How do you deal with missing values in the data? For example, not all patients have an active malignant disease. How does your clustering approach deals with that?

Answer: We handled missing values in principal component analysis (PCA), by imputing missing values in data sets using the PCA model. It was used as a preliminary step before performing PCA on the complete dataset. It is an iterative algorithm, consisting in (i) imputing missing values with initial values such as the mean of the variable, (ii) PCA is performed on the complete dataset, (iii) it imputes the missing values with the (regularized) fitted matrix. These three steps of estimation of the parameters via PCA and imputation of the missing values are iterated until convergence. Note that binary variables were also considered so that, for instance, patients free of malignant disease were coded as 0 for such variable. 

This has been detailed in the revised manuscript.

References

Josse, J. and Husson, F. missMDA (2016). A Package for Handling Missing Values in Multivariate Data Analysis. Journal of Statistical Software, 70 (1), pp 1-31

- Data normalisation is a must when applying PCA. What normalisation was applied to the data?

Answer: All data were scaled to unit variance

- While describing the clustering used, i.e., K-means, it is not clear how many components were computed and how many were used.

Answer: K-means used all the five computed components of the PCA. 

It was stated in the revised manuscript. 

- In the description of the results, it's not clear what causes of mortality are associated with each of the clusters. This brings me to think that K-means may not be the best approach to inform practitioners. Could be more useful in this case to consider a hierarchical clustering where you can identify subgroups similarity?

Answer: We agree that clustering is a subjective statistical analysis, and there is more than one appropriate algorithm for every dataset and type of problem. We do not think that this may rely on the causes of death given all the patients who died, died from COVID-19. One of alternative methods is hierarchical clustering where starting from one cluster, the algorithm splits depending on the similarities measured by the distance among points. We used hierarchical clustering, though it was not reported as such, and this has been highlighted in the revised manuscript. Revised Figure 2 has additionally plotted the dendogram resulting from the hierarchical clustering together with the results of the PCA, as illustrated below.

Impact and usability of the results:

- I agree with the authors that the results presented here are somehow expected considering the selection criteria of the cohort

Answer: Thank you

- My expectation of a paper like this is to present materials and methods along with the reproducible results that can be adopted in another institution. However, to that end, there are missing details to help others to use your results.

Answer: We have attempted to detail the report of the methods that would allow others to use the results.

- How can we test the reproducibility of your results? What should another hospital do to adopt your approach? What considerations should be taken? (e.g., this only works for patients above 48 years old)

Answer: We do agree that such clustering methods are data-driven do that external validity of our results are limited by essence. We mostly aimed to show the interests of focusing on defining patient patterns rather than focusing on covariates, separately. How this characterization could change management and prognosis should be evaluated in the future.

Questions that require an answer:

- How does the admission criteria to wards look like? Is there any standard followed?

Answer: The patient cohort is from the first wave of COVID 19. The criteria for hospitalization in a ward or intensive care unit could not be standardized at this time and they evolved over the study period according to management capacities. The patients analyzed were those with respiratory symptoms including those requiring oxygen supplementation (oxygen saturation at room air ≤ 94%). We have specified this point in the revised manuscript.

- Are patients with lost of the sense of smell considered in your analysis? This is based on various cases of asymptomatic patients that may not be considered in your study

Answer: Our cohort only included patients with COVID-19 admitted to the hospital wards due to respiratory symptoms of the disease. So, no patients were asymptomatic, and loss of smell was not a criteria for entry in the cohort. 

To get further insights in the influence of anosmia, we have reran the ACP additionally including this disease symptom. Results were rather unchanged, and this was explained by the correlation between digestive symptoms and anosmia, as displayed in the PCA plot below.

This has been included in the revised manuscript.

- The outcome results are extracted from the very early cases considered in the "first wave". I think it would be interesting to know if this has changed since then in the following waves (second, third, etc.)

Answer: We agree that this could be of interest. It has been reported in the revised Discussion section of the manuscript.

- Why the results were not compared against supervised models? I ask this because it should be very simple to perform this comparison.

Answer: Actually, we performed supervised models of survival based on Cox regression models. Results were tabulated in tables 3 and 4 of the manuscript. Those supervised analyses confirmed the prognostic value of variables used in the construction of the clusters, as previously reported. 

- [Line 270] "the poor outcome of patients with malignancies is worthy of attention" To me this is what deserves more attention and what could bring new insights into this study.

Answer: We thank the reviewer and agree. As mentioned in the introduction and discussion, our center is specialized in onco-hematology, which allows us to have a sufficient number of these patients to compare them to others. This is a specificity for our study.

**Minor comments**

Answer: All these points have been modified according to the Reviewer’s notes

- [Line 84] First mention of SARS-CoV-2, previously you refer only to COVID-19

- [Line 125] Typo, "Firs" → "First"

- [Line 125] "nonsupervised" → "non-supervised" Use hyphen

- [Line 133] Missing reference to the “NbClust” package

- [Line 134] Which are the best hyper-parameters found by the package?

- [Line 138] "nonparametric" → "non-parametric"

- [Line 144] p-values are denoted in different ways across the text, e.g., uppercase, lowercase, italic. Please chose one and stick to it.

- [Line 202] "Base line" → "Baseline"

- [Line 208] "mechanically" → "mechanical"

- [Line 215] "iQR" → "IQR" all in uppercase

- [Figure 1] It was not mentioned earlier in the text that patients whose stay was less than 2 days were left out of the analysis

- [Figure 3] What's the unit of time here?

 

Reviewer #4: In this paper, the authors analyze influence factors regarding the prognosis of a COVID-19 infection.

In general, the paper is well written and highly relevant to the current situation. However, a few points should be clarified.

- For me it is unclear how exactly the clustering is performed. In lines 131-132 you write that an iterative partitioning k-means method is used. This contradicts lines 133-135 where you say that the best clustering result is used (obtained by trying all combinations of the number of clusters, distance measures and clustering methods).

Answer: We apologize for the previous confusion in report.

Actually, clustering was based on K-means clustering that aims to partition the observations into k clusters in which each observation belongs to the cluster with the nearest mean (cluster centers or cluster centroid), minimizing within-cluster variance. It is also referred to as Lloyd's algorithm, particularly in the computer science community. Initialization used the Forgy method, that randomly chooses k observations from the dataset and uses these as the initial means. Several diagnostic checks were used, only for determining the number of clusters in the data set (while the packages also allowed to modify the method of clustering itself), and that which is selected more frequently by these indices is retained as the optimal number of clusters as proposed by Charrad et al (2014). Thus, based on those indices, 3 was the most frequently selected number of clusters, as detailed in the figure below. This has been clarified in the revised manuscript.

Reference

Charrad, Malika, Nadia Ghazzali, Véronique Boiteau, and Azam Niknafs. 2014. “NbClust: An R Package for Determining the Relevant Number of Clusters in a Data Set.” Journal of Statistical Software 61: 1–36

- In the PCA that your are doing, I think that you reduce the data-dimensions to two, this is not exactly described. What percentage of data is described by these two components?

Answer: The representation of the PCA indeed only concerned the two first components of the PCA (as displayed in figure 2), while the clustering was conducted on the five computed components of the PCA using the iterative partitioning k-means method. They summed up for 47.1% of the data variance. 

It has been more clearly stated in the revised manuscript.

- In line 156: Why exactly is 65 used as the age limit here? Can you present a histogram of the age distribution here?

Answer: We used 65 as the cutoff because it was close to the median (62.5), and similar to other studies

- The discussion of the results is rather brief and much is left to the reader to infer. Please expand the discussion of the results further.

Answer: We have expanded the Discussion section.

More Points

-------------

- Line 125: Firs_t_, we use ....

Answer: This has been corrected.

Recommendations

--------------

- I would like to suggest that autoencoder is used for dimensionality reduction, because it also finds nonlinear relationships in the data.

Answer: We agree that other learning method such as those based on neural networks, could have been used. We focused ourselves on the PCA given we were interested in the understanding of the profiles in terms of covariates, as illustrated for instance by their correlation. By contrast, neural networks such as autoencoders are used to automatically learn representative features from data, without explicitly relying on human assumptions. Moreover, the qualitative analysis of projections cannot be presented.

They have been reported and discussed in the revised manuscript.

- If you are looking for clusters that have high mortality, I would recommend the subgroup discovery method.

Answer: We agree that Subgroup discovery which is a data mining technique that extracts interesting rules with respect to a target variable, may have been of interest, as well as other semi-supervised learning methods. 

Nevertheless, it was not our aim. By contrast, we aimed at using only baseline information, that is at a time where no supervised target could be measured, to provide patient profile. It is only in a second step that we were interested in its potential relationship with the patient outcomes. This has been more clearly stated in the revised manuscript

---

## [Decision Letter · Decision Letter 1]

12 Apr 2021

Profiles and outcomes in patients with COVID-19 admitted to wards of a French oncohematological hospital: a clustering approach

PONE-D-21-02959R1

Dear Dr. Anne Bergeron,

We’re pleased to inform you that your manuscript has been judged scientifically suitable for publication and will be formally accepted for publication once it meets all outstanding technical requirements.

Kind regards,

Wisit Cheungpasitporn, MD, FACP

Academic Editor

PLOS ONE

Additional Editor Comments:

I reviewed the revised manuscript and the response to reviewers' comments. Revised Manuscript is well written. All comments have been addressed and thus accepted for publication.

Reviewers' comments:

Reviewer's Responses to Questions

**Comments to the Author**

1. If the authors have adequately addressed your comments raised in a previous round of review and you feel that this manuscript is now acceptable for publication, you may indicate that here to bypass the “Comments to the Author” section, enter your conflict of interest statement in the “Confidential to Editor” section, and submit your "Accept" recommendation.

Reviewer #2: All comments have been addressed

Reviewer #3: All comments have been addressed

2. Is the manuscript technically sound, and do the data support the conclusions?

Reviewer #2: Yes

Reviewer #3: Partly

3. Has the statistical analysis been performed appropriately and rigorously? 

Reviewer #2: Yes

Reviewer #3: Yes

4. Have the authors made all data underlying the findings in their manuscript fully available?

Reviewer #2: Yes

Reviewer #3: No

5. Is the manuscript presented in an intelligible fashion and written in standard English?

Reviewer #2: Yes

Reviewer #3: Yes

6. Review Comments to the Author

Reviewer #2: Looks good to me.

Reviewer #3: I would like to thank the authors for addressing all my comments from the first round. This revision, with the additional details on the methods, is clearer. I still believe there is much more research to be done with such type of data and using profiles such as the ones introduced here is a good place to start.

Because of that, I'd like to recommend this article for publication. Though I still have some minor comments, which I think they can be made while preparing the final submission.

Minor comments

------------------

- The quality of the images provided should be improved with a higher resolution

- The paper has a few different styles of writing. I'd recommend to proof read the document and ensure, for example, that you use either British or American english through the whole document and not a mix

- [Line 98] "from February 2020, to April 2020" → no comma needed

- [Line 119] "and the time to death was recorded" → replace "to" by "of"

- [Line 151] Missing reference for the Forgy method

- [Line 202] "likely to present with dyspnea" → remove "with"

- [Line 244] In terms of style, avoid parentheses within parentheses. You can use brackets inside the parentheses instead

- [Table 3] "DDimers" → "D-dimers"

- [Line 325] Missing reference for subgroup discovery

- In the conclusion section, there is a mention to a Spanish cohort, but there is no reference to the publication of that other analysis. Please add the missing reference.

7. PLOS authors have the option to publish the peer review history of their article (what does this mean?). If published, this will include your full peer review and any attached files.

Reviewer #2: No

Reviewer #3: **Yes: **Emir Munoz